# Estimation of Behavior Change Stage from Walking Information and Improvement of Walking Volume by Message Intervention

**DOI:** 10.3390/ijerph19031668

**Published:** 2022-02-01

**Authors:** Tomoya Yuasa, Fumiko Harada, Hiromitsu Shimakawa

**Affiliations:** 1Graduate School of Information Science and Engineering, Ritsumeikan University, Kusatsu, Shiga 525-8577, Japan; simakawa@cs.ritsumei.ac.jp; 2Connect Dot Ltd., Kyoto 604-0866, Japan; harada@de.is.ritsumei.ac.jp

**Keywords:** transtheoretical model, text messaging, exercise promotion, time-series analysis, Bayesian analysis

## Abstract

Lifestyle-related diseases are a major problem all over the world although exercising can prevent them. Therefore, it is necessary to encourage users to exercise regularly and to support their exercises. The purpose of this study is to investigate whether the estimation of behavior change stages can be predicted from the gait information obtained from wearable devices, and whether message interventions created based on the behavior change stages are effective in improving the amount of walking. As for the estimation of the behavior change stages, we investigated whether the behavior change stages could be correctly estimated compared with the ones obtained from the questionnaire. As for the effect of the message, we compared the period of no intervention with that of intervention to examine whether there was any change in the amount of walking. As a result of the experiment, we could not properly estimate the behavior change stage of users, but we found that the message intervention improved the amount of walking for many subjects. This suggests that further research is needed to estimate the stage of behavior change. However, message intervention is confirmed as an effective means to improve walking volume.

## 1. Introduction

Lifestyle-related diseases are a major problem around the world. The number of lifestyle-related diseases, such as diabetes and coronary artery disease, is increasing worldwide [1]. However, exercise can help reduce the risk of developing lifestyle-related diseases.

For example, a study by Park et al. has shown that a combination of aerobic and resistance exercise significantly improved body composition, arterial stiffness, and physical function in older obese men [2]. In addition, a study by RS Bafeberger Jr. et al. states that enduring vigorous sports activity, quitting smoking, maintaining normal blood, and avoiding obesity is individually associated with lower mortality from all causes and coronary heart disease [3].

Despite these many results showing that exercise is effective against lifestyle-related diseases, 1/4 of the world’s adults get hardly physically active [4]. There is an IT-based method to support people’s daily exercise. For example, nowadays, with the rapid spread of smartphones, many exercise support applications run on smartphones. These applications have many functions that provide self-monitoring and feedback for exercise. On the other hand, few functions provide information through health behaviors [5].

Therefore, we believe that it is a tool for people who are already interested in maintaining their health and are willing to exercise to continue exercising. However, not all people are motivated to exercise. To reduce the number of patients with lifestyle-related diseases, it is necessary to include not only those who exercise but also those who do not exercise and those who exercise irregularly. Increasing knowledge about disease and health behaviors is effective for people who belong to this group of people who are not interested in exercise [6].

Therefore, for people to make good use of existing exercise support applications, it is important to provide information that will trigger people to start or continue exercising. In addition, the market for smartwatches has been expanding in recent years, with global shipments of 266.3 million units in 2020 and expected to reach 776.23 million units by 2026 [7]. Therefore, smartwatches and applications can be used to personalize the number of steps taken in a day, etc. [8].

However, these proposals are made considering only the progress of the user’s physical activity and do not consider individual differences such as the user’s personality. For example, even when making the same suggestion, the situation differs between motivated and unmotivated users. Therefore, it is necessary not only to propose a personalized number of steps, but also to pay attention to the differences between users, and to provide more personalized support for each user. The Transtheoretical Model (TTM) [9] developed by Prochaska et al. is exercise support that reflects the differences among users. TTM is divided into five stages of behavioral change: indifference stage, interest stage, preparation stage, execution stage, and maintenance stage.

In fact, research using TTM has already been conducted in various fields [10,11]. For example, Arai et al. realized the enhancement of daily physical activity in college students by implementing a physical education class program corresponding to the behavior change stage [10]. However, most methods for estimating behavior change stages use questionnaires, which place a high burden on the user when measuring the user’s stage regularly. Therefore, to reduce the burden on the user, it is necessary to estimate the behavior change stage without using questionnaires. As a means of providing information to users, text message intervention is an effective means.

Research on text message intervention has already been conducted in various fields. Among them, several papers have shown that the Health Belief Model (HBM) [12] and Information-Motivation-Behavioral Skills Model (IMB) [13] can be used to improve exercise and motivation [14,15,16]. HBM is a model developed to explain health behavior; IMB conceptualizes the psychological determinants of behavior that may impair or improve health status. However, few studies have implemented long-term interventions with messages developed based on these models, considering the stages of behavior change, mainly among working adults.

A similar study was conducted by Adity et al. but the target population was limited to university students because text messages were used frequently [17]. However, the use of text messaging applications such as WhatsApp and LINE is not limited to any age group. Furthermore, college students are the least desirable age group in terms of health and lifestyle, while working adults are generally considered to have good health and lifestyle [18]. Therefore, we believe that the effectiveness of text messaging can be investigated more effectively by targeting working adults whose daily lives are more regular than university students whose daily lives are more disorganized. Based on the above, we believe that if we can estimate the stage of behavioral change from the user’s physical information and intervene with messages appropriate for that stage, it will lead to the improvement of the user’s exercise, which in the future will lead to the prevention of lifestyle-related diseases.

In this study, we conducted an experiment to verify the following two points:Whether intervention with messages based on the HBM model or the IMB model, which takes into account the behavior change stage, affects the number of steps.Can we estimate the user’s behavior change stage from the user’s gait information?

The paper answers these questions using the experiment.

## 2. Materials and Methods

### 2.1. Study Design

In this chapter, we propose to improve the amount of walking by considering the user’s busyness and behavior change stage. Walking is a form of exercise that is practiced by many people in countries around the world [19,20]. This suggests that walking is an exercise that anyone can easily engage in. Therefore, in this study, we focus on the amount of walking among the exercises. The outline of the proposed method is shown in Figure 1.

By adapting our method, users can receive messages appropriate to their behavior change stage and increase their daily walking volume without difficulty. First, we collect the user’s gait information from the wearable device. Next, we extract only the number of steps from the collected walking information, decompose the number of steps into a trend component and a seasonal component, and find the day of the week that promotes the user’s exercise from the seasonal component. In addition, we apply the Kalman filter to predict the user’s behavior change stage using the smoothed step count and other gait information. After that, we intervene with a message suitable for the user’s behavior change stage on the discovered days of the week to encourage walking. Since the user’s behavior change stage are variable, we can expect to improve the user’s walking volume with the periodical prediction of the behavior change stage based on walking information to send messages appropriate for the behavior change stage.

### 2.2. Participants

Thirteen male and female subjects (7 males and 6 females) in their 20s to 50s were tested. To collect subjects mainly from working adults, we recruited university staff and former students of the laboratory. In addition, there was one university student who was very interested in this experiment, so he participated as a subject. During the experiment, each subject wore a wearable device, Vivosmart 4 [21] by Garmin to record gait information. In this study, we collect gait information that can be collected from other wearable devices. The specific gait information is shown in Table 1.

### 2.3. Role of Duration

The duration of this experiment is shown in Figure 2.

Figure 2 (1) shows the “period to designate exercise promotion day”. (1) We extract seasonal components by STL decomposition [22] of the number of steps taken during the first four weeks. We summarize them by the average value for each day of the week. STL is an acronym for “Seasonal-Trend decomposition using Loess”. Loess is a method for estimating nonlinear relationships. By adapting STL decomposition, we can decompose time series data into trend components, seasonal components, and irregular components. The top two days of the week with the highest negative values are designated as “exercise promotion days” and intervention messages are given. The day of exercise promotion is set as a fixed day of the week during the period with message intervention, and each subject receives message intervention twice a week. (2) is the “period without intervention”. (3) and (4) are the “first half period with intervention” and the “second half period with intervention”. In this study, the period without intervention must be consecutive to the period with intervention to capture the number of steps in a time series. Therefore, the five days between (1) and (3) were used as the analysis period of the exercise promotion days. At the same time, since the subjects wore the wearable device and lived without the intervention by messages during this period, the period before the first half of (3) with intervention can be regarded as the period without the intervention by messages. Therefore, we can consider the period before the first half with intervention (3) as the period without message intervention. However, to have the same number of data for each period, we defined the four weeks before the first half period with intervention in (3) as the “no intervention period”.

Here, we intervene messages on the exercise promotion days obtained from the exercise promotion day identification period in (1). If the behavior change stage shifts from the indifference/interest period to the preparation/performance/maintenance period after the first half period with intervention in (3), the content of the intervening messages is switched from negative to positive messages in the second half period with intervention in (4). In the same way, if a person moves from the preparatory, implementation, or maintenance period to the indifference or interest period after the first half of the period with intervention in (3), the content of the message to be intervened is switched from a positive message to a negative message in the second half of the period with intervention in (4). The combination of (3) and (4) is called the “period with intervention” in (5).

Subjects will report their behavior change stage by answering a questionnaire once before the start of the experiment and every two weeks after the start of the experiment. The questionnaire was prepared by referring to the paper by Arai et al. [10] and rewriting the content to be related to this study.

Subjects were asked to select one item that applied to their behavior in response to the questions in Table 2, and the behavior change stage assigned to each item was defined as the subject’s current behavior change stage.

In addition, an Excel sheet with the dates during the experimental period will be distributed to the subjects in advance. If they exercise during the experimental period, they are required to describe the actual exercise they performed on the day of the experiment and the reason why they became motivated to exercise. Moreover, during the period of the message intervention, the subjects will answer a questionnaire about the message for each message that received the intervention.

### 2.4. Data Collection

In this study, we use a wearable device to acquire gait information. A wearable device can measure the number of steps more accurately than a smartphone [23]. For example, housework, such as cleaning and washing clothes, is a good form of exercise, and we can obtain a large number of steps. However, if you do housework with your smartphone on your desk, the pedometer on your smartphone will not count the number of steps you have taken. In addition, the smartphone is large enough to interfere with exercise, so it is possible to exercise at home. Since the purpose of this study is to improve the amount of daily walking, we do not want to lose the walking data from household chores. In the case of a wearable device, it is always worn and does not interfere with exercise. According to the World Health Organization (WHO), all physical activities are meaningful, including not only work and sports but also daily life activities [24]. Therefore, it is necessary to obtain the number of steps that can be obtained from detailed exercises as much as possible. In this study, we use a wearable device to acquire walking information.

### 2.5. STL Decomposition

By STL decomposition of the number of steps obtained from the wearable device and decomposing the number of steps into trend and seasonal components, we can make user-aware interventions. STL decomposition can be used not only for seasonal adjustment of monthly and quarterly data, but also for all kinds of time series data such as weekly and daily data. The seasonal component represents the cyclic number of steps taken by the user. In this study, we assumed that each week has a periodicity. In many countries around the world, we live in a cycle of one week, because of the application of the two-day workweek system. Therefore, the cycle with vacations is fixed.

In addition, since lecture times for students and regular meetings for working people are often fixed, there is a distinction between busy days and not-so-busy days for each day of the week. To extract the component of periodicity for each day of the week, the width of the component of the STL decomposition of the step count data was set to 7 days in this study. Mori et al. found that intervening messages on days of the week with negative values of the seasonal component increased the number of steps taken by users [25]. Therefore, there is a demand to examine the seasonal component for each user and to investigate the appropriate day of the week for intervention. Figure 3 shows an example of extracting only the seasonal component from the step count data and summarizing it as an average value for each day of the week.

If the value for each day of the week is positive, it means that the person usually walks on that day of the week. On the other hand, if each day of the week has a high negative value, it means that the person does not walk on that day of the week. In the case of Figure 3, Thursday and Friday are the days of the week when people walk more. On the other hand, Monday and Wednesday are the days of the week when people do not walk. On the other hand, the trend component obtained by STL decomposition is the number of unconscious steps taken by the user. Therefore, it is possible to see the number of steps a user takes every day.

Since the number of steps without STL decomposition contains irregular components, irregular elements such as an increase in the number of steps on the day due to an urgent errand are considered. Of course, it is desirable to temporally increase the number of steps due to higher motivation to exercise, but a temporary amount of walking is not effective in preventing lifestyle-related diseases. Therefore, users should increase the amount of unconscious usual walking. It is important to examine the user’s propensity components and observe the transition of the amount of walking.

### 2.6. Behavior Change Stage Estimation

To estimate the stage of behavioral change from the walking information, we use the walking information of the day, the sum and variance of the walking information for one week, the difference of the walking information for one week, and the percentage of the difference as explanatory variables. However, the number of steps was smoothed by applying a Kalman filter to reduce the effect of irregular fluctuations in the gait information. The Kalman filter can reduce the effect of irregular fluctuations while considering the seasonal and trend components of the gait volume.

In general, time-series data is represented by three basic components: trend, seasonal, and irregular. However, there must be other detailed components as well. Since it is unknown which components of the number of steps affect the estimation of the behavior change stage, the Kalman filter is applied instead of the STL decomposition to reduce the effect of irregular fluctuations while retaining the utilization of the fine components.

In addition, since this study considers one week as a cycle, we believe that comparisons for each week are also useful for estimating the behavior change stage. For example, if the number of steps per week is compared and there is an increasing trend in the number of steps compared to last week, we can guess that the participants exercised to gain steps during the week. If the percentage is high, the impact of one step is high, which suggests that the person is trying to move from a lower behavior change stage to the next stage. On the other hand, if the percentage is low, the impact of one step is low, which may indicate that the person is trying to move to a higher behavior change stage.

Furthermore, by using the variance over a week as an explanatory variable, we can check the upward and downward variation of the gait information over a week. For example, we consider that people who are in the lower or higher stages of behavior change do not exercise regularly or have a higher percentage of regular exercise, so we consider that the walking information is stable and the value of variance is small. On the other hand, for example, if a person is in the preparation period, he or she is likely to show interest in exercise and exercise irregularly. Based on the above, we adapt the differences, proportions, and variances of the gait information as explanatory variables. In addition, the behavior change stage obtained by the questionnaire was used as the objective variable. The behavior change stage is not known in what cycle the change occurs. However, assuming that the behavior change stage does not change in two weeks, the behavior change stage used as the objective variable was a fixed stage every two weeks. In this study, one subject was unable to complete the questionnaire to answer the behavior change stage once. However, since the results of the other questionnaires obtained did not show any change in the behavior change stage, the same behavior change stage was supplemented with the missing values. In this study, we used a random forest to estimate the target variable, the behavior change stage. The estimation method used in this study is shown in Figure 4. Random forests were run in Python 3.7.7 with sklearn.ensemble.RandomForestClassifier in scikit-learn 0.24.0.

To estimate the behavioral change stage without using questionnaires, it is necessary to estimate the behavior change stage of the subject to be estimated by using a model learned from the physical activity information and behavioral change stage data of other subjects. For this purpose, leave-one-out cross-validation is used as shown in Figure 4.

For 13 subjects, a model is constructed using the explanatory and objective variables of the remaining 12 subjects to estimate the behavior change stage of any one subject. By using any one explanatory variable as an input value to the constructed model, the estimated behavior change stage is obtained as an output value.

For example, to estimate the behavioral change stage of subject M, we first construct a model using the explanatory and objective variables of 12 subjects (A–L). By using all the explanatory variables of subject M as input values to this constructed model, the estimated behavior change stage is obtained as output.

Based on the gait information obtained from the wearable device (Table 1), (1) the weekly total(_total), (2) the difference of the total from last week(_diff), (3) the ratio of last week(_rate), and (4) the weekly variance was calculated and used as explanatory variables(_var), respectively. Therefore, to process each of the eight types of gait information in four different ways, a total of 32 explanatory variables were used.

The list of explanatory variables is shown in Table 3.

Before _, the type of walking information is shown, and after _, the processing method of walking information is shown. For example, “Number of steps_total” represents “total number of steps for one week”. In addition, “Total calories_var” represents “dispersion of calories burned in one week”.

### 2.7. Composing Text Messages

We intervene with messages based on the Health Belief Model (HBM) [12] and the Information-Motivation-Behavioral Skills Model (IMB) [13]. Messages are created based on these models, and the messages are divided into two types: positive messages and negative messages. Behavior change stages are divided into five stages, and it is often said that cognitive processes are adapted to the indifference and interest stages, and behavioral processes are adapted to the preparation, execution, and maintenance stages [26]. Cognitive processes include experiencing a range of emotions associated with the risk of continuing an unhealthy behavior. Therefore, during the indifference and interest periods, we intervene with negative messages describing the possible risks of not exercising. Behavioral processes include keeping things around that are associated with changing and maintaining healthful behaviors. Therefore, during the preparation, execution, and maintenance periods, positive messages describing the benefits of exercise should be used as interventions.

To add diversity to the content of the messages, we asked nine men and women in their 20s (8 men and 1 woman) to help us compose the messages. The authors were asked to create a message to intervene with the subjects to include one of the constituent contents of the HBM and IMB models. Examples of messages for the HBM and IMB models are shown in Table 4 and Table 5. This time, the IMB model “Behavior information” was not created, so it is not adapted.

## 3. Results

### 3.1. Changes in the Number of Steps Due to Message Intervention

To understand the impact of the intervention by messages created based on the HBM and IMB models that consider the behavior change stage, only the trend component is extracted by STL decomposition of the number of steps taken during the experimental period for each subject. Subject G was mistakenly sent a message that was not suitable for the behavior change stage only once during the message intervention, so the effects of this mistake are included. The number of steps of the extracted trend component for subject A is shown in Figure 5.

The blue line in Figure 5 shows the number of steps extracted by STL decomposition during the whole experiment. The period before the red line in Figure 5 is the period without intervention, the period between the red and green lines is the first half period with intervention, and the period after the green line is the second half period with intervention. The results in Figure 5 show that the number of steps increased immediately after the intervention of the message, and at the end of the experiment, the number of steps increased more than the maximum value during the period without the intervention of the message. Since the number of steps tends to increase after the message intervention for other subjects as well, it is necessary to display the median number of steps for each subject in each period to quantitatively see how much the number of steps increases or decreases due to the message intervention. Therefore, we extracted the trend components by STL decomposition of the number of steps in the period without intervention, the first half period with intervention, the second half period with intervention, and the period with intervention for each subject, and the median values of each are shown in Table 6.

Since there were cases where we could not obtain gait information from several subjects during the experiment, we applied the STL decomposition by supplementing the missing values during the period without intervention with the average values of each day before the missing values occurred, and the missing values during the period with intervention with the average values of each day during the period without intervention. When the tendency component obtained by STL decomposition was visualized using a boxplot, some subjects recognized that part of the step count values extracted as the tendency component was outliers, so the effect of the outliers was small. Therefore, the median value was adopted.

The results in Table 6 show that most subjects increased the number of steps during the period with the intervention compared to the period without message intervention. Among the periods with intervention, the median value of the trend component of the number of steps was higher in the second half period with intervention than in the first half period with intervention. The Shapiro-Wilk test, which is a method to test whether the data are normally distributed, was conducted assuming that the population of data sampled for the null hypothesis follows a normal distribution. The Shapiro-Wilk test is a method to test whether the data is normally distributed or not, and the test cannot be conducted because there are cases where the null hypothesis is rejected or not depending on the period. However, the results in Table 6 show that intervention with messages created based on the HBM and IMB models, which consider the stages of behavior change, led to an increase in the number of steps in many subjects. Therefore, it can be considered that the message intervention may affect the increase in the number of steps.

A study by Adity et al. [16] showed that the number of steps increased more in the second half of the period with intervention than in the first half of the period with intervention. As a cause, it is suggested that the change in behavior due to the message intervention did not occur immediately, but appeared gradually over time. In fact, in this study, when we checked the subjects’ excerpts concerning their exercise contents and motivation to exercise, they remembered past text messages such as “I remembered the exercise messages on 7/21 and 7/23 and exercised”. This suggests that text messages can be accumulated to promote one’s own exercise over time. It also implies that some of these messages can be remembered and become triggers to perform exercises.

### 3.2. Short-Term Effect of the Message

The results of 3.1 suggest that long-term intervention with messages based on the HBM and IMB models, which consider the behavior change stage, is likely to have an impact on increasing the number of steps. Next, we investigated whether exercise was promoted immediately after the actual intervention of the message. In this study, we applied Bayesian Online Change Point Detection (BOCD) [27], which is a change point extraction method using a Bayesian framework, to extract the part of the step count of each subject that changed rapidly. Figure 6 shows the results for subject D, to whom BOCD was applied. If a change is extracted on the day of the message intervention, it is considered that the number of steps increased rapidly after receiving the message.

The blue line in Figure 6 shows the transition of the actual number of steps during the experiment. It is difficult to extract the change points from the transition of the number of steps of the trend component by STL decomposition. It is also hard to extract the change points from the transition of the number of steps smoothed by the Kalman filter. The change of the number of steps is smooth in the time series produced with them. We used the actual number of steps in BOCD. The circled points in the data are the change points detected by BOCD. To examine the influence of the message intervention on these change points, we checked them by comparing them with the Excel sheet containing the date of the message intervention and the content of the exercise.

As a result, it was found that the detected changes could be divided into three major categories: those that were the result of the exercise performed on the day of the message, those that were performed within a short period after receiving the message, and those that were unrelated to the message. This indicates that the effects of the message intervention are not limited to the day of receiving the message but extend over multiple days. A short-term increase in the number of steps is not considered effective in preventing lifestyle-related diseases. However, if users experience exercise to increase their step count even for a short period, it will be the first step to making exercise a habit. Therefore, the intervention of regular messages can be considered a tool to provide a trigger for the user to do so.

### 3.3. Estimating Behavior Change Stage Based on Walking Information

The leave-one-out cross-validation shown in Figure 4 was conducted, and the estimated behavior change stages for each subject were concatenated. In addition, the actual behavior change stages for each subject obtained from the questionnaire on behavior change stages were concatenated. The overall rate of correct answers was calculated by comparing them, and the result was 0.65.

In this experiment, the behavior change stage did not change, and many subjects were always in the interest stage. Therefore, we believe that we have constructed a strong model that captures the estimation results as the interest stage. Therefore, while we were able to predict the interest stage, we were not able to guess other stages of behavioral change, such as the preparation stage and the execution stage, and this is why we got such a rate of correct answers.

We believe that this is due to the bias of the subjects. Looking at the behavior change stages of the subjects who cooperated in this experiment, 8 out of 13 subjects, or more than half of the total, were in the interest stage. Furthermore, from the results of the questionnaire regarding the stage of behavioral change at the end of the case period, we found that 6 out of 13 subjects were in the interest stage, and many subjects shifted from the interest stage to the preparation stage and did not settle down. The reason for this is that there was a lot of data from the interest stage, and it was not possible to obtain information on other stages such as the preparation stage and the implementation stage.

## 4. Discussion

### 4.1. Valid Message Types

During the experiment, the subjects were given 16 messages twice a week each for 8 weeks with intervention. By examining which messages influenced each subject, we can understand which messages resonate with users in improving the number of steps for health promotion. Based on McCoy et al.’s study [28], subjects were asked

whether the messages they received made them feel as if they were themselveswhether the messages increased their knowledge about physical activitywhether the messages increased their confidence that they could increase their physical activityDid the messages help you increase your physical activityDid the messages change your awareness of physical activity

Each of the five items was evaluated on a five-point scale. The evaluation scores for each item were totaled, and the message with the highest total value was considered to be the message that had the greatest impact on each subject.

As a result, it was found that several messages had a common impact on each subject. The results showed that several messages had a common influence on each subject. Therefore, we considered that certain components of the messages influenced the improvement of the number of steps taken by the subjects. The results are shown in Table 7 and Table 8.

Table 7 and Table 8 suggest that the inclusion of the perception of severity component can be a trigger for users to increase their steps.

### 4.2. Validity of Variables in Behavior Change Stage

To investigate the variables that affect the estimation of each subject’s behavior change stage, I visualized the importance of the variables to each subject’s estimation model in the leave-one-out cross-validation conducted in Section 3.3. For the visualization, I used a method called feature_importances_ included in sklearn.ensemble. The feature_importances_ is obtained by calculating the amount of decrease in the criterion called Gini impurity for each explanatory variable, normalizing them, and adding them up. The larger the decrease, the more important the variable is. As a result of the visualization, the top three variable importance levels for each subject are shown in Table 9.

From Table 9, it can be seen that Weekly exercise_total, The number of steps_total, Weekly exercise_var have an effect on the estimation of the behavior change stage for many subjects. Weekly exercise_total is the sum of the time when the step rate or heart rate exceeded the threshold for appropriate exercise for at least 10 min from the day to the week before, Number of steps_total is the sum of the number of steps from the day to the week before, and Weekly exercise_var is the variance of time when the step rate or heart rate exceeded the threshold for appropriate exercise for at least 10 min from the day to the week before. Furthermore, the visualization results showed that the values of the key variables of Weekly exercise_total and Number of steps_total are particularly strong, and the difference with Weekly exercise_var is large. Therefore, it can be seen that the behavior change stage is judged based on the amount of walking and the amount of exercise on the day to estimate the behavior change stage in this study. This indicates that the number of steps taken every day is an important variable for estimating the stage of behavioral change.

In fact, regarding the relationship between the transformation stage of walking behavior and walking time, the later the transformation stage of walking behavior, the longer the walking time [29]. However, in this study, variables other than the amount of walking and physical activity on the day are considered to have a small effect on the behavior change stage, so if the number of steps and physical activity is high, the person is judged to be in the maintenance stage, while if they are low, the person is judged to be in the indifference stage. However, even if the number of steps taken is the same, the stage of behavioral change is naturally different between those who consciously exercise for health and those who are not interested in exercise but walk a certain number of steps every day due to work or other reasons. Therefore, we think that the explanatory variables used in this study are not effective for estimating the stage of behavioral change.

## 5. Conclusions

As a result of this study, it was found that the intervention with messages created based on the HBM model and IMB model considering the behavior change stage had an impact on the number of steps taken. As an effect of the message intervention, the number of steps increased more in the latter half of the message intervention period than in the whole period. This suggests that some of the messages may be remembered and triggered by the accumulation of messages to encourage exercise. It was confirmed that the intervention of all the messages affected the actual amount of walking even in the short term. In addition, we were not able to estimate the stage of behavioral change of the user from the user’s walking information. This may be because there was a bias in the behavior change stage of the subjects and the explanatory variables for predicting the behavior change stage were not properly selected. Therefore, in the future, it is necessary to increase the number of subjects and equalize the distribution of the behavior change stages of the subjects. In this study, the number of subjects was limited because we used wearable devices of the same product prepared in advance to eliminate the measurement error of gait information caused by using different wearable devices. I believe that additional subjects can be more easily added to the study because the range of devices can be expanded by providing additional identical products or by using other wearable devices from Garmin. In addition, further discussion of explanatory variables for predicting the appropriate behavior change stage from the user’s walking information will make the estimation of the behavior change stage more appropriate.

## Figures and Tables

**Figure 1 ijerph-19-01668-f001:**
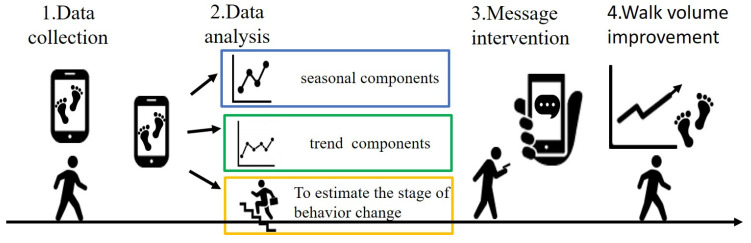
A schematic diagram of the proposed method.

**Figure 2 ijerph-19-01668-f002:**
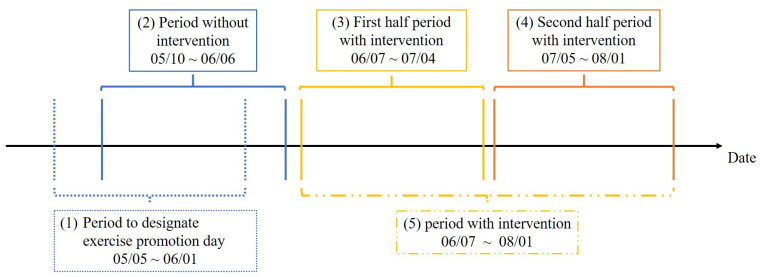
The duration of this experiment.

**Figure 3 ijerph-19-01668-f003:**
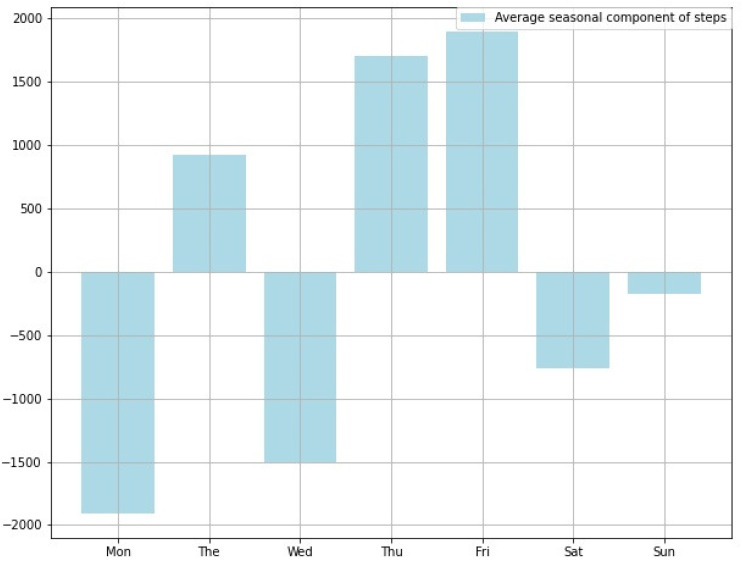
Average seasonal component of steps.

**Figure 4 ijerph-19-01668-f004:**
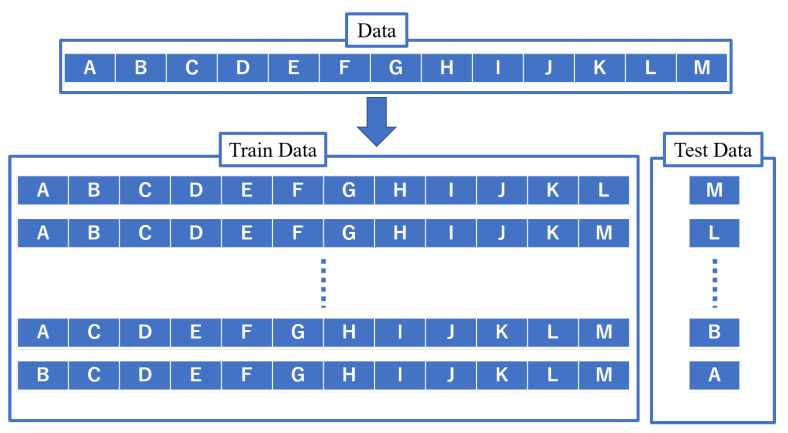
Behavior change stage estimation method.

**Figure 5 ijerph-19-01668-f005:**
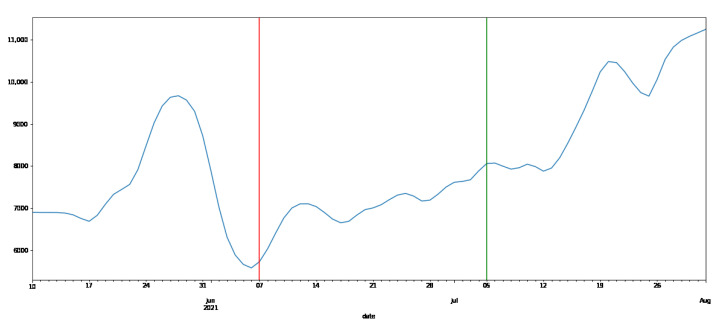
Trend component of steps.

**Figure 6 ijerph-19-01668-f006:**
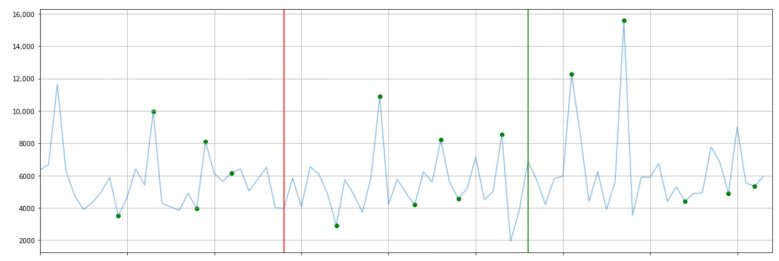
BOCD.

**Table 1 ijerph-19-01668-t001:** The specific gait information.

No.	Collection Data	Detailed Information on Collection Data
1	Number of steps	Total number of steps taken in a day
2	Total calories	Calories burned during exercise in a day
3	Stress	Average heart rate variability in a day
4	Weekly exercise	At least 10 min of exercise in which the step rate or heart rate exceeds the threshold of what is considered an appropriate exercise in a day
5	Upward stairs	Number of stairs climbed in a day
6	Downward stairs	Number of stairs descended in a day
7	Up and down the stairs	Total number of stairs climbed or descended in a day
8	Heart rate	Average heart rate per day

**Table 2 ijerph-19-01668-t002:** Questions about behavior change stages.

Q: What is Your Closest Position?	Behavior Change Stage
A: Have no interest in preventing lifestyle-related diseases through exercise, and have no plans to do so.	Indifference stage
A: There is interest in preventing lifestyle-related diseases through exercise, but there are no immediate plans to do so.	Interest stage
A: I am interested in the prevention of lifestyle-related diseases through exercise, and I practice it occasionally.	Preparation stage
A: Recently, I have been exercising to prevent lifestyle-related diseases.	Execution stage
A: I have been exercising for a long time to prevent lifestyle-related diseases.	Maintenance stage

**Table 3 ijerph-19-01668-t003:** Explanatory variables.

No.	Explanatory Variable	No.	Explanatory Variable
1	Number of steps_total	17	Number of steps_rate
2	Total calories_total	18	Total calories_rate
3	Stress_total	19	Stress_rate
4	Weekly exercise_total	20	Weekly exercise_rate
5	Upward stairs_total	21	Upward stairs_rate
6	Downward stairs_total	22	Downward stairs_rate
7	Up and down the stairs_total	23	Up and down the stairs_rate
8	Heart rate_total	24	Heart rate_rate
9	Number of steps_diff	25	Number of steps_var
10	Total calories_diff	26	Total calories_var
11	Stress_diff	27	Stress_var
12	Weekly exercise_diff	28	Weekly exercise_var
13	Upward stairs_diff	29	Upward stairs_var
14	Downward stairs_diff	30	Downward stairs_var
15	Up and down the stairs_diff	31	Up and down the stairs_var
16	Heart rate_diff	32	Heart rate_var

**Table 4 ijerph-19-01668-t004:** About HBM.

Components of the HBM	Message Example
Perceived susceptibility	Lifestyle-related diseases account for about 30% of medical expenses in Japan and about 50% of the death toll. Therefore, it is a disease that anyone is likely to get if not prevented. However, it is possible to prevent lifestyle-related diseases by exercising on a daily basis! First of all, let’s walk every day, aiming for him to walk as much as one step from yesterday !!
Perceived severity	Did you know that the calf is called the “second heart”? Weakness in the calf causes it to weaken its function as a pump that circulates blood. The circulation of blood, water, lymph, etc. will be impaired, causing various physical disorders such as swelling and coldness, so walk to prevent muscle weakness!
Perceived benefits	According to the Centers for Disease Control (CDC), regular exercise has more benefits than exercise-related injuries, such as reducing depression and anxiety and reducing the risk of premature death. It is said that there are many. I think it’s difficult to do a lot of exercises suddenly, so let’s start exercising from today with the consciousness of “increasing physical activity and exercise habits more than now”!
Perceived barriers	Because exercise is important, why not exercise hard to move efficiently in a short period of time? Sudden strenuous exercise or heavy physical exertion increases the risk of death. There is evidence that sudden and intense physical activity is proportional to mortality. It is important to maintain moderate exercise every day!
Self-efficacy	Lack of exercise increases the risk of developing lifestyle-related diseases. To prevent lifestyle-related diseases, it is important to have exercise habits. “Walking” is one of the familiar exercises. All you need is clothes and shoes, so let’s walk a lot.
Cue to action	According to a research survey conducted in Nakanojo Town, Gunma Prefecture, it is possible to prevent bedridden by walking about 2000 steps (1400 m). Let’s start with what we can do, such as walking to one station.

**Table 5 ijerph-19-01668-t005:** About IMB.

Components of the IMB	Message Example
Behavior motivation	Are you staying at home in Corona and have less chance of being exposed to sunlight? Sunlight on the skin may lead to a deficiency of vitamin D synthesized in the body. It promotes calcium absorption and muscle synthesis and regulates and maintains immune function, so if it is insufficient, it will not be possible to maintain a healthy body. Why don’t you leave the house and take a short walk to get the sun?
Behavior skills	How long do you walk every day? The average number of steps taken by the Japanese is 6793 for men and 5832 for females. If you haven’t reached the average number of steps, try walking 10 min more than usual. In fact, just walking for 10 min will increase the number of steps by about 1000 steps. If you haven’t reached the average number of steps, why not take a detour today and go home with a different view?

**Table 6 ijerph-19-01668-t006:** Results of steps for each subject.

Subjects	Period without Intervention.	First Half Period with Intervention	Second Half Period with Intervention.	Period with Intervention
A	15,583	17,827	15,282	16,617
B	14,480	14,967	13,688	14,906
C	9779	9648	11,226	9882
D	5486	5456	6034	5763
E	10,399	9107	10,740	10,256
F	6559	6271	5975	6057
G	3842	4118	5039	4713
H	7354	5609	4373	4865
I	6110	6045	6128	6114
J	6109	5961	6254	6152
K	6001	5930	5620	5846
L	11,240	12,353	13,664	13,170
M	7055	7089	9701	7878

**Table 7 ijerph-19-01668-t007:** Valid types of HBM models.

Components of the HBM	Count
Perceived susceptibility	3
Perceived severity	6
Perceived benefits	2
Perceived barriers	1
Self-efficacy	3
Cue to action	4

**Table 8 ijerph-19-01668-t008:** Valid types of IMB models.

Components of the IMB	Count
Behavior motivation	5
Behavior skills	4

**Table 9 ijerph-19-01668-t009:** Important variables in behavior modification stage estimation.

Subjects	1st Place	2nd Place	3rd Place
A	Weekly exercise_total	Number of steps_total	Weekly exercise_var
B	Weekly exercise_total	Weekly exercise_var	Number of steps_total
C	Number of steps_total	Weekly exercise_total	Weekly exercise_var
D	Number of steps_total	Weekly exercise_total	Weekly exercise_var
E	Weekly exercise_total	Number of steps_total	Weekly exercise_var
F	Number of steps_total	Upward stairs_total	Upward stairs_var
G	Number of steps_total	Weekly exercise_total	Weekly exercise_var
H	Weekly exercise_total	Number of steps_total	Weekly exercise_var
I	Number of steps_total	Weekly exercise_total	Weekly exercise_var
J	Number of steps_total	Weekly exercise_total	Weekly exercise_var
K	Weekly exercise_total	Number of steps_total	Weekly exercise_var
L	Number of steps_total	Weekly exercise_total	Weekly exercise_var
M	Weekly exercise_total	Number of steps_total	Weekly exercise_var

## Data Availability

The data presented in this study are available on request from the corresponding author. The data are not publicly available due to privacy.

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
