# Peer review of "Estimation of Behavior Change Stage from Walking Information and Improvement of Walking Volume by Message Intervention"

_ijerph, 2022, doi:10.3390/ijerph19031668_

Round 1
Reviewer 1 Report
The authors have used an intervention experiment to demonstrate whether the behavior change stage can be estimated. I have a few questions about the methods and the results.
- In line 66, the reference for the Health Belief Model wasn't correctly added which showed as a question mark.
- In line 121, please spell out STL.
- In line 149, please include the questions related to the behavior change stage from the questionnaire in the appendix and explain how the value was determined based on subjects' answers.
- In line 243-245, the same sentence was shown repeatedly.
- In line 246, the authors mentioned busing the algorithm random forester. Do you mean "random forest"? Figure 4 seems like the leave-one-out cross-validation was also used. Please be specific about all methods used and properly cite them. In addition, please include the software and any libraries/packages used to run the algorithms.
- In line 251, what does it mean by "using any one explanatory variable as an input value to the constructed model"? Does it mean by fitting all explanatory variables into the model? It is not clear how many explanatory variables were fitted in total. I highly recommend listing all of them with their definitions in a table rather than using 3-4 paragraphs.
- It seems to me that Section 4.1 are still discussing results that should belong to the results section.
- In line 330, it is not clear how the accuracy was calculated for each subject. Also, if leave-one-out cross-validation was used, it is recommended to calculate an overall accuracy by concatenating all predicted values rather than reporting accuracy for each subject.
- In line 359, there is no reference for Bayesian Online Change Point Detection.
- In line 315, no statistical tests were conducted to support the conclusion that intervention with messages is likely to have an impact. We can't just draw a conclusion just based on eyeballing the results in Table 4.
- In Line 406, please describe the metric used to rank the variable importance and how it was calculated.
Author Response
Thank you very much for your important comments. I appreciate the time and effort you put into them. My response to the referee's comments is in pdf format.
Please see the attachment.

Reviewer 2 Report
I am recommending minor revision for this paper. I believe it is an interesting topic to the readers. I only have a few comments to the author.
-
In Line 66, there is a missing reference.
-
Should cite STL decomposition method.
“Cleveland, R. B., Cleveland, W. S., McRae, J. E., & Terpenning, I. J. (1990). STL: A seasonal-trend decomposition procedure based on loess. Journal of Official Statistics, 6(1), 3–33.” -
Can the author add more details description about what is STL decomposition? It will help the reader to understand the paper easily.
-
Can the author provide a justification about why 13 patients in this study or how they determined the number of patients? Can the study include more patients?
Author Response

(The authors gave the same response as above.)

Round 2
Reviewer 1 Report
Thank you for taking the time to address all my comments. I don't have any further questions.